# Nanopore-based targeted next-generation sequencing (tNGS): A versatile technology specialized in detecting low bacterial load clinical specimens

Chen Yang[1][☯], Weiwei Gao[2][☯], Yicheng Guo[1], Yi Zeng[1]*

1 Department of Tuberculosis, The School of Public Health of Nanjing Medical University, The Second Hospital of Nanjing, Nanjing, China, 2 Department of Tuberculosis, The Second Hospital of Nanjing, Nanjing, China

☯ These authors contributed equally to this work.

* 960559051@qq.com

## Abstract

### Introduction

The 2024 global tuberculosis report indicated that the epidemiological situation of tuberculosis remains concerning. Current tuberculosis detection methods have limitations, highlighting the urgent need to develop more convenient, effective, and widely utilized detection technologies in clinical settings to facilitate early diagnosis and treatment guidance for tuberculosis. Nanopore-based targeted next-generation sequencing (tNGS) offers advantages such as convenience, efficiency, and long-read sequencing, making it a commonly used method for mycobacteria identification.

### Methods

This study compared the detection efficiency of tNGS with Xpert MTB/RIF, MTB culture, and AFB smear in sputum samples, BALF samples, and pathological tissue samples to evaluate the clinical applicability of tNGS in diagnosing tuberculosis and low bacterial load tuberculosis, including extrapulmonary and smear-negative cases.

### Results

Among the four detection methods, tNGS demonstrated the highest efficiency, sensitivity, specificity, and AUC values, which are 93.4%, 94.7%, and 0.94, respectively. This method was particularly advantageous for detecting tuberculosis in patients with low bacterial loads, as evidenced by a significantly different positive detection rate in histopathological specimens compared to other methods (P < 0.001). Furthermore, tNGS achieved overall positive detection rates of 93.8% for smear-negative tuberculosis patients and 89.1% for culture-negative tuberculosis patients, both of which were significantly higher than those of other detection methods (P < 0.05).

**Data availability statement:** The original data are deposited to the National Center for Biotechnology Information (NCBI, accession number PRJNA1238507), link: https://www.ncbi.nlm.nih.gov/sra/?term=PRJNA1238507. The Minimal Data Set for this study can be found in the Supporting Information.

**Funding:** This study is supported by Nanjing Health Science and Technology Development Special Fund [grant number: M2021073]. The funders had no role in study design, data collection and analysis, decision to publish, or preparation of the manuscript.

**Competing interests:** The authors have declared that no competing interests exist.

Additionally, tNGS could directly identify bacterial strains and detect mutations associated with drug resistance. In this study, the agreement rate between tNGS strain identification of NTM and the final diagnosis was 94.1%. Among the 21 identified mutation sites associated with rifampicin resistance, one (Pro454His) was located outside RRDR.

## Conclusion

It is anticipated that tNGS will play a crucial clinical role in the early prevention and control of tuberculosis in the future.

---

## 1. Introduction

Tuberculosis(TB), caused by Mycobacterium tuberculosis(MTB), is a serious contagious disease predominantly affecting the lungs(pulmonary tuberculosis, PTB) but can also infect other parts of the body(extrapulmonary tuberculosis, EPTB). According to the World Health Organization(WHO) global tuberculosis report in 2024 [1], 10.8 million new cases of tuberculosis were estimated in 2023, a slight increase from 10.7 million cases in 2022. Furthermore, tuberculosis would kill 1.25 million people globally by 2023, making it the world's leading cause of death from a single infectious disease. The global epidemiological situation for tuberculosis is critical. More concerning is that current methods for detecting Mycobacterium tuberculosis(MTB) have limitations that may lead to inadequate diagnosis of TB patients and continue to increase the burden on society and patients [2].

The acid-fast bacteria (AFB) staining smear method, a traditional technique for identifying TB, is straightforward and quick but lacks sensitivity [3] and cannot differentiate between MTB and non-tuberculosis mycobacteria(NTM) [4]. While the culture method is considered the most reliable for detecting MTB, it is time-consuming and requires high biosafety measures [5], which hinders early disease detection. Despite advancements in molecular biology technology improving MTB diagnosis, common detection methods still have drawbacks [6]. For instance, GeneXpert MTB/RIF(Xpert), recommended for the primary diagnosis of PTB by WHO [7], is effective in detecting smear-positive pulmonary tuberculosis but is not ideal for specimens with low bacterial loads (like smear-negative PTB [8] and EPTB [9]), which limits its widespread use [10]. Besides, since Xpert is capable of detecting dead MTB [11], it may yield false positive results when screening individuals who have undergone TB treatment in the past or have a history of TB. Thus, to further improve the diagnostic capabilities for TB and address the shortcomings of current detection methods, we should improve the convenience, efficiency, and precision of TB detection technologies. [6,12].

Nanopore-based targeted next-generation sequencing(tNGS), a novel third-generation technology, offers fast sequencing speed, long-read length, and real-time data monitoring [13]. This technology uses nanoprotein pores, or nanopores, as biosensors. They are embedded in a resistive polymer membrane and create an ionic current through the nanopores by applying a steady voltage to the electrolyte solution. Under the motor

protein's traction, the negatively charged single-stranded DNA flows through the nanopores. The sensor will record the changes in various ionic currents brought on by various bases during the translocation process, and a specific algorithm will be used to obtain the nucleotide sequence to perform quick and real-time bioinformatics analysis [14]. With the widespread use of tNGS in infectious disease detection and human genomics research [15,16], there is growing support for this new technology to play an important role in the identification of mycobacteria and the detection of resistance genes [17–19].

Previous research primarily employed tNGS to analyze respiratory samples for assessing the diagnostic effectiveness of pulmonary tuberculosis. However, it appears that the significant potential of tNGS has not been adequately explored. The key technical advantage of tNGS lies in its ability to specifically target and enrich the nucleic acid of MTB in clinical samples through specialized capture technology, thereby yielding more comprehensive data for detection. This suggests that tNGS can offer a pathogenic basis for the clinical diagnosis of tuberculosis with low bacterial loads, including smear-negative tuberculosis and extrapulmonary tuberculosis. Thus, our study not only focused on respiratory specimens but also incorporated a variety of tissue samples (such as those from the skin, lymph nodes, and pleura) to preliminarily investigate the application potential of tNGS and assess its diagnostic efficacy in cases like smear-negative pulmonary tuberculosis. In addition, tNGS can directly identify bacterial strains in clinical specimens, providing better diagnostic and treatment opportunities for other mycobacterial patients. We aim to provide a viable direction for clinical researchers working in this field.

## 2. Materials and methods

### 2.1 Study design

This research was conducted at the Tuberculosis Department of the Second Hospital of Nanjing, Jiangsu Province, China. The study involved a retrospective analysis of clinical data from suspected tuberculosis patients who had visited the department between January 2023 and January 2024. Medical professionals collected various samples from patients based on their specific health conditions. If patients could not provide adequate or qualified sputum samples, it was recommended that BALF samples be collected. In cases where the nature of the lesion was unclear, making a differential diagnosis difficult, or when there were clinical indications of extrapulmonary lesions, it was advised to collect pathological tissue samples. We ensured that specimen collection respected the interests and privacy of the providers. Informed consent was obtained from patients and their families before sample collection, with all patients signing consent forms. In addition, all patients in the study underwent targeted next-generation sequencing (tNGS), GeneXpert MTB/RIF, Mycobacterium tuberculosis (MTB) culture, and acid-fast bacilli (AFB) smear tests simultaneously. Patients who did not meet the study criteria were excluded. The study's workflow is illustrated in Fig 1. The data for this research was sourced exclusively from the inpatient records of the Second Hospital of Nanjing. After gathering comprehensive and qualifying sample data, it would be verified and anonymized before being entered and organized in Microsoft Excel 2010 for further statistical analysis. All data intended for research had been collected by May 12, 2024.

Referring to the most authoritative health industry standards in China, including the Diagnostic Standards for Pulmonary Tuberculosis (WS288–2017), the Classification Standards for Tuberculosis (WS196–2017), and the Guidelines for the Diagnosis and Treatment of Nontuberculous Mycobacteriosis (2020) [20], clinical experts would use clinical manifestations, medical imaging characteristics, pathogenic examination results, and responses to general antituberculosis treatment to determine the final clinical diagnosis for all patients. For tuberculosis patients, this involved detecting Mycobacterium tuberculosis (MTB) in respiratory specimens or histopathological samples, positive outcomes from molecular biology tests like Gene Xpert MTB/RIF, or immunological tests such as interferon-gamma release assay (IGRA), along with the presence of clinical symptoms, signs, imaging findings, and other pertinent auxiliary test results. For NTM patients, criteria included positive NTM isolation and culture results, positive bacterial identification results for NTM like the direct homologous gene or sequence comparison method [21], and relevant clinical, imaging, and pathological examination findings related to NTM disease.

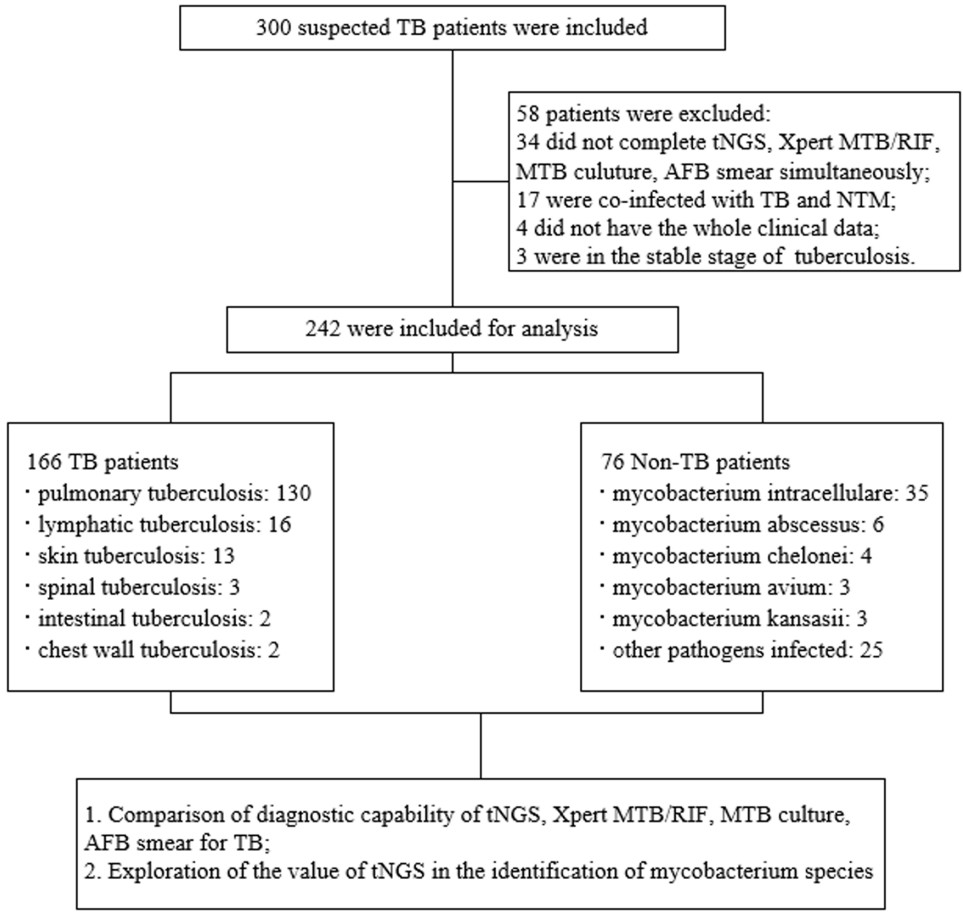

**Fig 1. Flowchart illustrating the categorization of patients involved in the current research.**

## 2.2 Ethics

Informed consent was obtained from patients and their families before sample collection. All participants provided written informed consent in their primary language. This study received approval from the Human Investigation Ethics and System Review Committee of Nanjing Second Hospital[ID:2024-LS-ky026).

## 2.3 Specimen collection

In this research, we utilized the patient's fresh sputum, alveolar lavage fluid, and pathological tissue samples for relevant testing. The sputum specimens referred to the morning sputum coughed up by the patient upon waking. The broncho-alveolar lavage fluid(BALF) specimens were obtained by lavaging the affected lung area during bronchoscopy. The pathological tissue specimens were tissues collected through puncture, biopsy, and surgical resection.

## 2.4 Acid-fast bacilli (AFB) smear and Mycobacterium tuberculosis (MTB) culture

According to the 2022 update of the Practical Manual on Tuberculosis Laboratory Strengthening [22] by the World Health Organization (WHO), this research rigorously followed the laboratory procedures for testing tuberculosis. Acid-fast staining microscopy involves creating a smear from a patient's sample, staining it with a dye like Auramine O, heating the smear, and then examining it under a microscope to check for the presence of mycobacteria. MTB culture involved inoculating the

collected material onto either a solid medium (Lowenstein-Jensen medium) or a liquid medium (BACTECTM MGITTM960 system) to observe the growth of MTB.

## 2.5 GeneXpert MTB/RIF

Initially, the sample was mixed thoroughly with a sample handling solution. Next, the mixture was incubated at room temperature for 15 minutes before being transferred to the GeneXpert MTB/RIF kit (Cepheid, USA) using the sterile dropper provided. The mixture was then loaded into the test module of the GeneXpert instrument (Cepheid, USA) for the automated detection of MTB. Subsequently, the system will automatically display the MTB detection result within a 2-hour.

## 2.6 Nanopore-based targeted next-generation sequencing(tNGS)

Initially, the patient samples were liquefied, and the top liquid layer was removed. The sediment was then mixed with a lysis solution to ensure the even resuspension of each particle. DNA extraction and purification were conducted using the QIAamp DNA Microbiome kit (QIAGEN, Canada) and quantified with a Qubit fluorometer 4.0 (Thermo Fisher Scientific, MA) following the provided instructions. The purified products were barcoded according to the Native Barcoding Expansion 1–12 guidelines (Oxford Nanopore Technologies, UK). DNA amplification was performed using PCR on an ABI 2720 Thermal Cycler (Applied Biosystems, USA). Genomic DNA libraries were constructed using the DNA ligation kit SQK-LSK110 (Oxford Nanopore Technologies, UK) and the GridION X5 platform (Oxford Nanopore Technologies, UK), following the manufacturer's protocols. Real-time sequencing data acquisition was conducted using MinKnow software (Oxford Nanopore Technologies, UK). The obtained sequencing data underwent quality filtering and the removal of duplicate sequences to align with the reference sequences. Human DNA reads were filtered out by aligning them to the human reference genome (GRCh38) using Minimap2 software [23]. The remaining filtered reads were aligned against the MTB reference sequence (NC_000962.3) and Mycobacterium sequence (txid 1763) to identify Mycobacterium tuberculosis and Non-Tuberculous Mycobacteria (NTM) within 48 hours, enabling direct strain identification of NTM species.

## 2.7 Threshold parameters and decision framework for tNGS results

In this study, we considered a sample positive for tuberculosis if the reads per million (rpm) for the tuberculosis sequence exceed 1000. However, we acknowledged the potential influence of sequencing depth on diagnostic classification. Therefore, in our analysis, the average depth of coverage for the samples ranged from 90x to 1000x, with a median of 315x, which we believe is sufficiently high for accurate diagnosis. A detailed tNGS result decision framework is as follows:

(1) Quality Control Steps to Minimize Errors:

① Base-Calling Algorithms: We utilized the latest base-calling algorithm provided by Oxford Nanopore Technologies, designed to improve the accuracy of base identification. The Guppy base caller was used with the high-accuracy model to maximize the precision of base calls.

② Read Filtering: Low-quality reads (length < 200nt or >1000nt, or Q-score < 7) were removed.

③ Sequence Alignment: We used minimap2, a splice-aware aligner, to map our reads to a reference genome. This tool is optimized for long reads and is effective in identifying and filtering out reads with misalignments or structural variations that could lead to errors in variant calling.

(2) Handling Low-Quality Reads:

Low-quality reads were filtered out using a combination of read length and quality score thresholds. Specifically, reads shorter than 200nt or longer than 2000 nt, or with a mean Phred quality score (Q-score) below 7 were excluded from the analysis. This filtering step ensured that only high-quality reads were used for downstream analyses.

(3)      Result Determination

Mycobacterium tuberculosis is deemed positive if the RPM exceeds 1000. For drug resistance, we generally require: for regions associated with drug resistance mutations, a sequencing depth greater than 10x, with a minimum depth of 5x for the specific mutation sites, and the mutation frequency must be at least 10%. Only when all three conditions are met do we report drug resistance.

## 2.8  Statistical analysis

In this research, we conducted the statistical analysis using R software (version 4.4.1). The diagnostic accuracy of four detection methods (tNGS, Xpert MTB/RIF, MTB culture, and AFB smear) was assessed based on the final diagnosis. Sensitivity, specificity, positive predictive value (PPV), negative predictive value (NPV), and the area under the curve (AUC) value were calculated. The positive detection rates of tNGS were compared with those of other methods using paired chi-square tests on various clinical specimens from tuberculosis patients. Fisher's exact probability method and chi-square tests were used to compare the positive detection rates of tNGS and other methods for patients with smear-negative and culture-negative pulmonary tuberculosis. In addition, ROC curves and half-box diagrams were employed to illustrate the diagnostic efficacy of tNGS in tuberculosis, particularly among patients with low bacterial loads. The Venn diagram and Upset diagram were utilized to demonstrate the intersections between different assays yielding positive test results in both enrolled patients and TB patients. The Sunburst chart was used to present rifampicin-resistant mutations on the gene rpoB identified by tNGS

## 3.  Results

### 3.1  Clinical information of the study participants

In this research, we initially examined electronic case records of 300 suspected tuberculosis patients and eliminated 58 individuals based on specific criteria: 1) 34 patients did not undergo all tests simultaneously; 2) 17 patients were co-infected with both MTB and NTM; 3) The clinical information of 4 patients was incomplete; 4) Three patients were in a stable tuberculosis stage. Ultimately, 242 patients were part of the study, with 166 diagnosed with tuberculosis: 130 with pulmonary tuberculosis and 36 with extrapulmonary tuberculosis. Additionally, 76 patients had non-tuberculosis conditions, 51 had NTM disease, and 25 had infections from other pathogens (Fig 1). Each patient contributed only one sample, totaling 31 sputum samples, 116 BALF samples, and 95 pathological tissue samples. None of the patients tested positive for HIV. The demographic details and laboratory test results of the participants included in the study are presented in Table 1.

### 3.2  Comparison of diagnostic efficiency of tNGS, Xpert MTB/RIF, MTB culture, and AFB smear

Adhere to the established diagnostic criteria for tuberculosis in China, which encompass clinical data, laboratory test results, and imaging findings, to determine whether the research participant is afflicted with tuberculosis. Utilize this evaluation as a benchmark to assess the diagnostic efficacy of all testing methods. In the diagnosis of tuberculosis, the sensitivity, specificity, positive predictive value, negative predictive value and AUC value of tNGS were 93.4, 94.7, 97.5, 86.7, and 0.94 respectively; The values of Xpert MTB/RIF were 69.9, 92.1, 95.1, 58.3, and 0.81, respectively; The values of MTB culture were 51.8, 98.7, 98.9, 48.4, and 0.75, respectively; The values of AFB smear were 35.5, 76.3, 76.6, 35.2, and 0.56, respectively. All results are summarized in Table 2. The ROC curves of the four tests for tuberculosis detection are displayed in Fig 2.

### 3.3  The distribution and overlap of positive results from tNGS, Xpert MTB/RIF, MTB culture, and AFB smear

In cases where other detection methods yielded negative results, targeted next-generation sequencing (tNGS) could independently detect 32 positive tests, acid-fast bacilli(AFB) smear could independently detect 20 positive tests, Xpert MTB/RIF could independently detect 8 positive tests, while Mycobacterium tuberculosis (MTB) culture failed to detect any

 

**Table 1. Characteristics of the included patients.**

| Characteristic | Overall(n = 242) | TB(n = 166) | Non-TB(n = 76) | P-value |
|---|---|---|---|---|
| Sociodemographic information | | | | |
| Age | 49.0 [33.0,65.0] | 58.0 [48.0,69.0] | 43.0 [29.0,59.8] | <0.001 |
| BMI | 20.5 [18.4,24.2] | 21.9 [19.5,24.2] | 19.6 [17.8,23.3] | 0.003 |
| Gender | | | | 0.007 |
| Male | 122(50.4) | 94(56.6) | 28(36.8) | |
| Female | 120(49.6) | 72(43.4) | 48(63.2) | |
| Region | | | | 0.318 |
| City | 134(55.4) | 96(57.8) | 38(50.0) | |
| Countryside | 108(44.6) | 70(42.2) | 38(50.0) | |
| Clinical features | | | | |
| Fever | | | | 0.392 |
| Yes | 63(26.0) | 40(24.1) | 23(30.3) | |
| No | 179(74.0) | 126(75.9) | 53(69.7) | |
| Cough | | | | 0.262 |
| Yes | 151(62.4) | 108(65.1) | 43(56.6) | |
| No | 91(37.6) | 58(34.9) | 33(43.4) | |
| Night sweat | | | | 0.788 |
| Yes | 72(26.8) | 48(28.9) | 24(31.6) | |
| No | 170(73.2) | 118(71.1) | 52(68.4) | |
| Weight loss | | | | 0.087 |
| Yes | 30(12.4) | 16(9.6) | 14(18.4) | |
| No | 212(87.6) | 150(90.4) | 62(81.6) | |
| Laboratory results | | | | 0.080 |
| Leukocyte | 5.66 [4.71,6.98] | 5.66 [4.84, 6.86] | 5.61 [4.09, 7.12] | 0.417 |
| Erythrocyte | 4.30 [3.94,4.76] | 4.41 [3.98, 4.81] | 4.22 [3.71, 4.53] | 0.030 |
| Neutrophils | 3.54 [2.63,4.58] | 3.53 [2.70, 4.34] | 3.68 [2.39, 5.12] | 0.377 |
| Lymphocyte | 1.52 [1.16,1.98] | 1.48 [1.11, 2.02] | 1.58 [1.23, 1.85] | 0.805 |
| Gamma-interferon release test | | | | 0.376 |
| Positive | 104(43.0) | 75(45.2) | 29(38.2) | |
| Negative | 138(57.0) | 89(54.8) | 47(61.8) | |

positive tests (Fig 3. a). In addition, we conducted a comparison between tNGS and other detection methods in identifying tuberculosis patients through the Venn diagram (Fig 3. b). The findings revealed that tNGS shared 30 positive results with Xpert, 32 positive results with both Xpert and MTB culture, and 41 positive results with the three detection methods.

### 3.4 Comparison of positive detection rate between tNGS and Xpert MTB/RIF, MTB culture, AFB smear for tuberculosis

In this research, 166 tuberculosis patients were included, with 20 identified through sputum samples, 78 through BALF samples, and 68 through pathological tissue samples. Overall, there was a significant difference in the tuberculosis detection rate between tNGS and the other three methods (P<0.001). Additionally, when comparing the tuberculosis detection rates of tNGS and other methods across different sample types, significant differences were observed in the detection rates of tuberculosis in pathological tissues and BALF samples (P<0.05). In sputum samples, apart from AFB smear, there was no significant distinction in the tuberculosis detection rates between tNGS and Xpert, MTB culture (P=0.231). These findings are summarized in Table 3.

**Table 2. Diagnostic efficiency of the four tests for tuberculosis.**

| Test | Sensitivity(%,95%CI) | Specificity(%,95%CI) | PPV(%,95%CI) | NPV(%,95%CI) | AUC(value,95%CI) |
|---|---|---|---|---|---|
| tNGS | 93.4(88.2-96.5) | 94.7(86.4-98.3) | 97.5(93.3-99.2) | 86.7(77.1-92.9) | 0.94(0.91-0.97) |
| Xpert MTB/RIF | 69.9(62.2-76.6) | 92.1(83.0-96.7) | 95.1(89.2-98.0) | 58.3(49.0-67.2) | 0.81(0.76-0.86) |
| MTB culture | 51.8(44.0-59.6) | 98.7(91.9-99.9) | 98.9(92.9-99.9) | 48.4(40.3-56.5) | 0.75(0.71-0.79) |
| AFB smear | 35.5(28.4-43.4) | 76.3(64.9-85.0) | 76.6(65.3-85.2) | 35.2(28.0-43.0) | 0.56(0.50-0.62) |

PPV: positive predictive value; NPV: negative predictive value; AUC: area under the curve; MTB: Mycobacterium tuberculosis; AFB: acid-fast bacilli.

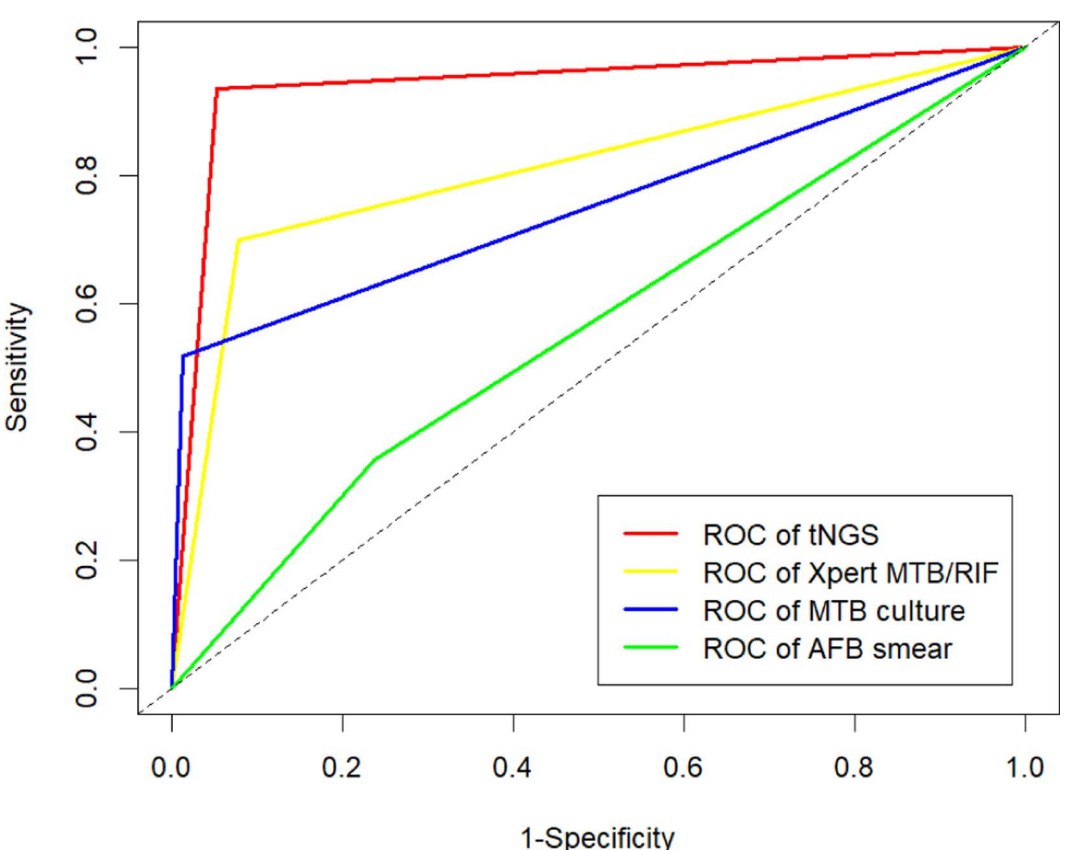

**Fig 2. ROC curves of the four tests for tuberculosis.**

### 3.5 Comparison of positive detection rate between tNGS and Xpert MTB/RIF, MTB culture, AFB smear for smear-negative pulmonary tuberculosis

In this study, we examined 81 patients with smear-negative tuberculosis, which included 7 patients with sputum samples, 47 patients with BALF samples, and 27 patients with histopathological samples. Additionally, we identified 55 patients with culture-negative tuberculosis, comprising 3 patients with sputum samples, 26 patients with BALF samples, and 26 patients with histopathological samples. Tables 4 and 5 present the positive detection rates of tNGS, Xpert, and MTB culture for patients with smear-negative tuberculosis, as well as the positive detection rates of tNGS, Xpert, and AFB smear

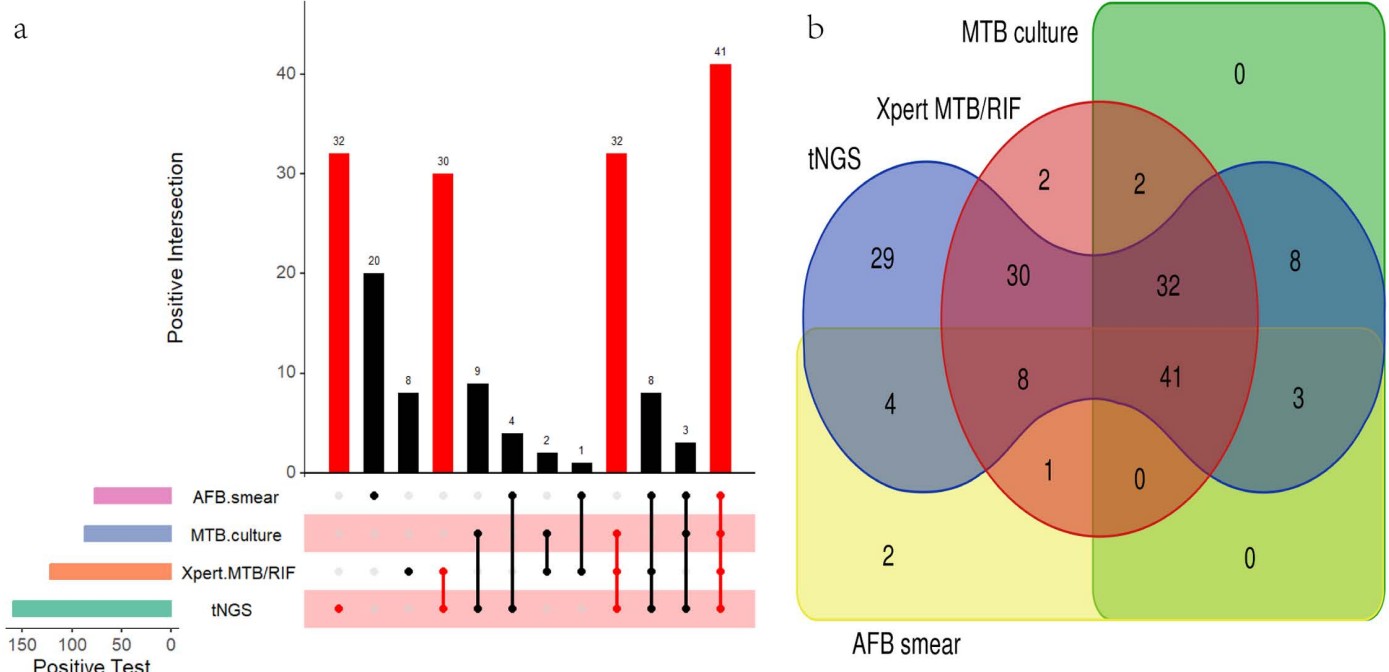

**Fig 3. The scenarios of positive outcomes from various detection methods.** (a) Upset diagram of positive tests for the included patients. (b) Venn diagram of positive tests for tuberculosis patients.

for patients with culture-negative tuberculosis. For patients with smear-negative pulmonary tuberculosis, the positive detection rates for tNGS in sputum, BALF, and histopathological specimens were 100.0%, 100.0%, and 81.5%, respectively, resulting in a total positive detection rate that surpassed other methods (93.8% vs. 71.6% and 42.0%, p < 0.001). In the case of culture-negative tuberculosis patients, the positive detection rates for tNGS in sputum, BALF, and histopathological specimens were 100.0%, 100.0%, and 76.9%, respectively, leading to a total positive detection rate that also exceeded other methods (89.1% vs. 63.6% and 14.5%, p < 0.05).

### 3.6 Nucleotide sequence numbers tNGS detected in tuberculosis patients

The nucleotide sequence number in clinical samples can be directly identified by tNGS to assess the bacterial content in various specimens. In tuberculosis cases, the median nucleotide sequence number detected by tNGS in pathological tissue samples, BALF samples, and sputum samples was 27, 132, and 1105.5, respectively. A graphical representation in the form of a half-box line diagram (Fig 4) illustrates the distribution of nucleotide sequence numbers detected by tNGS in various clinical specimens from patients with TB.

### 3.7 Evaluation of tNGS detection of rifampicin resistance patterns

The rifampicin resistance-related mutations identified by tNGS were all found on the rpoB gene(Fig 5). Of the 21 rifampicin-resistant mutations, 20 were located in the rifampicin resistance-determining region (RRDR): 11 were Ser531Leu, 2 were His526Tyr, 2 were His526Leu, 1 was Ser512Gly, 1 was Asp516Val, 1 was Leu524fs, 1 was Gln531Pro, and 1 was Leu-533Pro. Additionally, one resistance mutation (Pro454His) was identified outside the RRDR of the rpoB gene, underscoring a significant advantage of tNGS as a long-read sequencing method.

**Table 3. Comparison of positive detection rate between tNGS and the other three tests for tuberculosis.**

| Comparison | Sputum | | BALF | | Tissue | | Overall | |
|---|---|---|---|---|---|---|---|---|
| | $x^2$ | P-value | $x^2$ | P-value | $x^2$ | P-value | $x^2$ | P-value |
| tNGS vs Xpert MTB/RIF | 1.441 | 0.231 | 7.546 | 0.006 | 21.566 | <0.001 | 29.001 | <0.001 |
| tNGS vs MTB culture | 1.441 | 0.231 | 28.846 | <0.001 | 56.953 | <0.001 | 70.000 | <0.001 |
| tNGS vs AFB smear | 6.234 | 0.008 | 64.434 | <0.001 | 65.027 | <0.001 | 118.656 | <0.001 |

**Table 4. Comparison of positive detection rate between tNGS, Xpert, and MTB culture for smear-negative pulmonary tuberculosis.**

| Test | Sputum | | BALF | | Tissue | | Overall | |
|---|---|---|---|---|---|---|---|---|
| | Rate(%) | P-value | Rate(%) | P-value | Rate(%) | P-value | Rate(%) | P-value |
| tNGS | 100(7/7) | / | 100(47/47) | / | 81.5(22/27) | / | 93.8(76/81) | / |
| Xpert | 71.4(5/7) | 0.462 | 85.1(40/47) | 0.018 | 48.1(13/27) | 0.023 | 71.6(58/81) | <0.001 |
| MTB culture | 71.4(5/7) | 0.462 | 51.1(24/47) | <0.001 | 18.5(5/27) | <0.001 | 42.0(34/81) | <0.001 |

**Table 5. Comparison of positive detection rate between tNGS, Xpert, and AFB smear for culture-negative pulmonary tuberculosis.**

| Test | Sputum | | BALF | | Tissue | | Overall | |
|---|---|---|---|---|---|---|---|---|
| | Rate(%) | P-value | Rate(%) | P-value | Rate(%) | P-value | Rate(%) | P-value |
| tNGS | 100(3/3) | / | 100(26/26) | / | 76.9(20/26) | / | 89.1(49/55) | / |
| Xpert | 66.7(2/3) | / | 80.8(21/26) | 0.051 | 48,1(12/26) | 0.046 | 63.6(35/55) | 0.004 |
| AFB smear | 33.3(1/3) | / | 11.5(3/26) | <0.001 | 15.4(4/26) | <0.001 | 14.5(8/55) | <0.001 |

## 4. Discussion

Nanopore-based targeted next-generation sequencing (tNGS) can play a key role in the early containment of tuberculosis transmission. Our research findings demonstrated that tNGS was more effective in detecting tuberculosis compared to other methods. The sensitivity, specificity, and AUC values of tNGS were 93.4%, 94.7%, and 0.94 respectively, which were significantly better than the other detection methods. This was consistent with the research results of Yu et al [24]: they used nanopore sequencing to detect respiratory specimens from 164 suspected tuberculosis patients, with sensitivity, specificity, and AUC values of 94.8%, 97.9%, and 0.96, respectively. Notably, tNGS identified 44 more cases of tuberculosis compared to Xpert, including 25 cases of pulmonary tuberculosis, 9 cases of skin tuberculosis, 7 cases of lymph node tuberculosis, 2 cases of intestinal tuberculosis, and 1 case of pyramidal tuberculosis. In addition, we investigated the ability of all detection methods in detecting positive results independently. When results from other detection methods were negative, tNGS showed a positive predictive value of 90.6% (29/32), identifying four types of tuberculosis: 16 cases of pulmonary tuberculosis, 7 cases of skin tuberculosis, 4 cases of lymph node tuberculosis, and 2 cases of intestinal tuberculosis. AFB smear had a positive predictive value of 10% (2/20), with 1 case of pulmonary tuberculosis, 1 case of skin tuberculosis, and the remaining 18 cases diagnosed as NTM. Xpert MTB/RIF demonstrated a positive predictive value of 25% (2/8), all of which were diagnosed as pulmonary tuberculosis. Obviously, tNGS outperformed others in terms of both quantities and types when independently detecting tuberculosis cases.

Pulmonary tuberculosis is the most common form of tuberculosis and is the primary source of tuberculosis transmission. It can be understood that previous studies had predominantly focused on utilizing tNGS to identify respiratory samples and assess their diagnostic significance in pulmonary tuberculosis [24–26]. However, we appear to be paying insufficient attention to extrapulmonary tuberculosis, even though it has posed a significant burden on public health in China [27]. Yang et al. [28] used tNGS to detect cerebrospinal fluid samples from 122 suspected intracranial tuberculosis patients. The results showed that the sensitivity, specificity, and AUC values of tNGS were 60.0%, 95.5%, and 0.78,

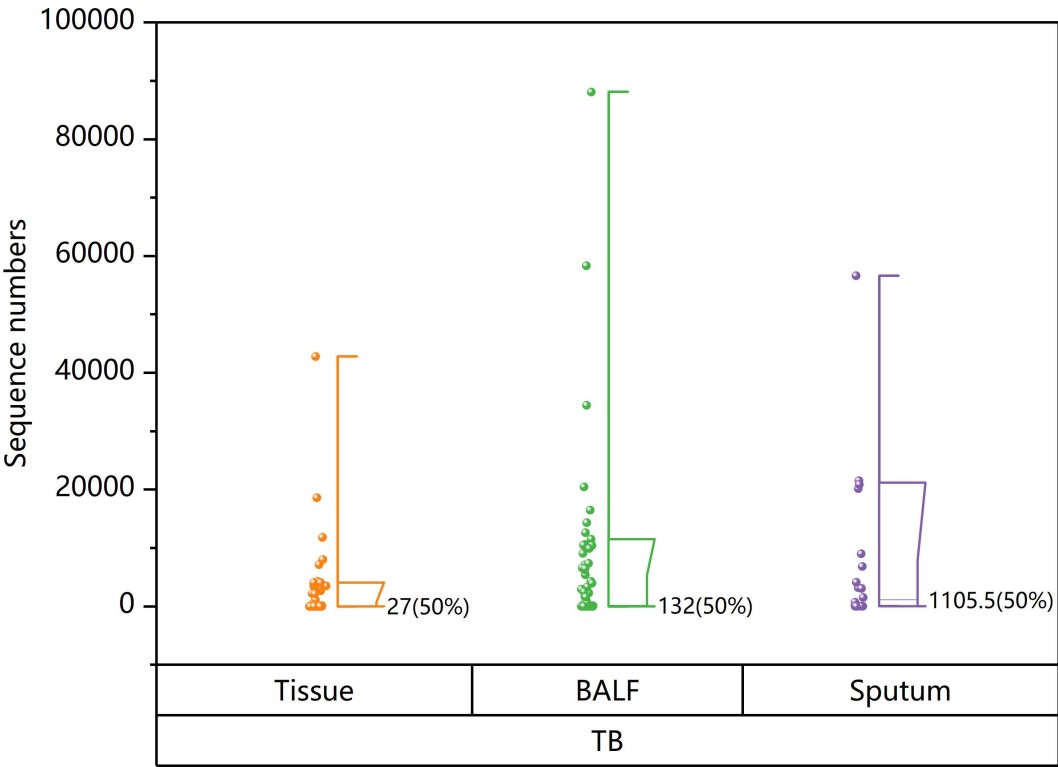

**Fig 4. Half-box diagram of the sequence numbers tNGS detected in TB patients.**

respectively. Although the overall performance decreased compared to detecting respiratory specimens, it was still significantly better than other detection methods. Addressing how to provide reliable laboratory evidence for extrapulmonary tuberculosis patients will be a crucial area of focus in the future.

In our research, we found that the effectiveness of tNGS in detecting tuberculosis was significantly better than other detection methods overall (P < 0.001). In particular, tNGS exhibited unique advantages (P < 0.001) over other detection methods for pathological histological specimens. This aligned with the findings of Gao et al [19], who employed tNGS to analyze 110 histopathological samples, with a sensitivity of 88.2% and a specificity of 94.1%, significantly better than other detection methods. Histopathological specimens used to diagnose extrapulmonary tuberculosis are frequently thought to have a low bacterial load, which presents significant challenges to traditional detection methods [29,30]. However, tNGS can enrich MTB nucleic acids in clinical specimens through specialized capture techniques and gather adequate data for detection [31]. This explained why tNGS had such a significant positive detection rate in pathological histological specimens.

The bacterial content of different specimens can be evaluated by using tNGS to directly identify the nucleotide sequence number in clinical samples. The median number of tuberculosis sequences detected by tNGS in sputum samples was found to be 1105.5(Fig 4). This suggested that the sputum samples from tuberculosis patients in the study contained a significant amount of bacterial content and masked the performance disparities among detection methods(P > 0.05). In contrast, pathological tissue samples and BALF samples in this research typically exhibited lower bacterial presence with a median of 27 and 132 tuberculosis sequences, respectively. Nevertheless, the positive detection rates of tNGS for tuberculosis in tissue and BALF samples were significantly better than that of other methods(P < 0.05), highlighting the diagnostic efficacy of tNGS in tuberculosis patients with low bacterial loads. Interestingly, among the 11

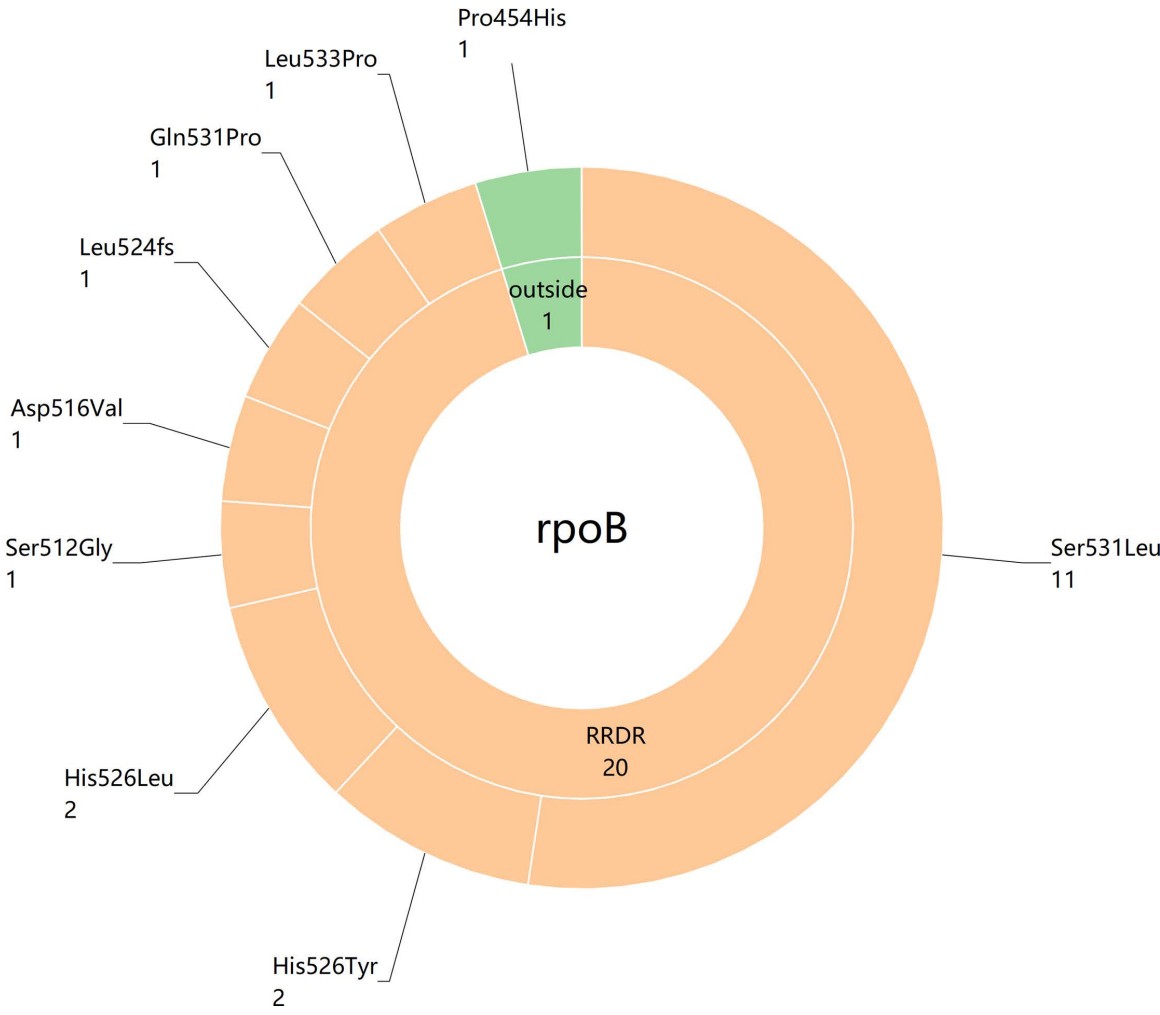

**Fig 5. Sunburst chart displaying the resistance mutations on the rifampicin-resistant gene identified by tNGS.**

tuberculosis patients who were not detected by tNGS in this study, 5 patients were discovered by Xpert, 2 patients were discovered by MTB culture, and 3 patients were discovered by AFB smear, all of which were histopathological specimens without exception, and most of them were lung biopsy tissues. However, there is currently no research evaluating the differences in the detection of tNGS between lung biopsy specimens and respiratory specimens, and further exploration is needed.

Improving the detection rate of smear-negative pulmonary tuberculosis patients has been a persistent challenge in clinical practice [32,33]. While Xpert is known for its relatively high positive detection rate for pulmonary tuberculosis, studies have shown varying efficiency levels, ranging from 57.1% to 96.3% [34–36]. This variability may be attributed to smear-negative pulmonary tuberculosis often being categorized as having low bacterial loads. In this study, Xpert demonstrated a 71.6% positive detection rate for smear-negative pulmonary tuberculosis, with the highest rate of 85.1% in BALF samples. However, the lowest positive detection rate of 48.1% was observed in pathological tissue samples. In contrast, tNGS demonstrated a higher positive detection rate of 93.8% for smear-negative pulmonary tuberculosis, similar to Yu G et al.'s findings [37] (91.7%). Notably, tNGS achieved an 81.5% positive detection rate for smear-negative pulmonary tuberculosis in pathological tissue specimens, significantly outperforming Xpert (P = 0.023). This study also suggested that

BALF samples might be the most effective specimen (with the highest positive detection rate) for diagnosing smear-negative pulmonary tuberculosis, which was consistent with previous research [36]. We anticipate that enhancing the positive detection rate for smear-negative tuberculosis through tNGS in the future will contribute to the continued progress in tuberculosis prevention and control efforts.

Currently, the positive isolation rate of NTM is increasing, making NTM disease a growing public health concern [20]. Traditional detection methods are unable to directly discern the subgroups and subspecies of NTM, necessitating further bacterial identification. Our study explored the accuracy of tNGS in detecting NTM, revealing a high consistency rate of 94.1% (48/51) with the final diagnosis. Errors in reporting could be attributed to tNGS misidentifying Mycobacterium chelonae subsp. abscessus as Mycobacterium abscessus. Akram SM et al. [38] argue that the identical gene sequences of Mycobacterium chelonae and Mycobacterium abscessus in the 54–510 region resulted in their classification as the same species for a considerable period. However, the detection of the intergene sequence (ITS) and hsp65 gene now allows for their differentiation. Therefore, we inferred that the reason for the misreporting of tNGS in this study might be due to confusion in the sequence regions of the 54–510 gene. Similarly, Huang YY et al. [39] employed nanopore-based targeted sequencing to promptly identify a patient with Mycobacterium marinum, overcoming the drawbacks of lengthy NTM culture periods. Notably, while PCR currently cannot differentiate between Mycobacterium marinum and Mycobacterium ulcerans [40], tNGS has proven effective in identifying Mycobacterium species. It is anticipated that tNGS will play a crucial role in diagnosing infectious diseases caused by mycobacteria, various bacteria, fungi, and other pathogens in the future [41].

We also evaluated the consistency between tNGS, Xpert, and phenotypic drug sensitivity testing(pDST) in identifying tuberculosis patients with rifampicin-resistant. Among the 166 tuberculosis patients included in this study, 22 were found to be resistant to rifampicin through pDST. Notably, all detected clinical specimens were respiratory samples, and we did not identify any rifampicin-resistant patients from histopathological specimens. In this study, the consistency rate for detecting rifampicin resistance using tNGS was 95.6% (21 out of 22), compared to a consistency rate of 68.2% (15 out of 22) for Xpert MTB/RIF. These findings closely aligned with the research conducted by Liu et al. [17], which reported a detection sensitivity for rifampicin using tNGS of approximately 93.5%, while the sensitivity of Xpert was at 83.8%. Notably, tNGS could detect mutations outside of the RRDR of the rpoB gene, like Pro454His in this study(Fig 5), whereas Xpert MTB/RIF failed to do so. Furthermore, tNGS is emerging as a trustworthy technique for providing comprehensive drug-resistance information for drug-resistant TB patients. Colman et al. [42] and Wu et al. [43] both used tNGS to evaluate the drug resistance of more than ten kinds of anti-tuberculosis drugs. The results showed that the sensitivity and specificity of this method for most drugs remained above 90%, showing good diagnostic performance. Significantly, tNGS identified mutations linked to resistance against crucial novel and repurposed drugs (such as bedaquiline and linezolid), which were currently undetectable by any other WHO-recommended rapid diagnostic tests available on the market [42]. In the future, tNGS is likely to become an alternative method of phenotypic drug sensitivity test(pDST) to assist clinicians in developing more accurate personalized treatments for drug-resistant TB patients.

Furthermore, tNGS is recognized as a cost-effective approach for diagnosing TB. Numerous studies are currently investigating the application of tNGS in identifying Mycobacterium and detecting drug-resistance genes. Ye et al. [44] conducted research on the comprehensive detection of respiratory pathogens in BALF from patients with pulmonary infections using tNGS. Their findings demonstrated that compared to traditional methods, tNGS achieved a significantly higher positive detection rate for suspected tuberculosis cases (87.18% vs. 48.72%) and exhibited superior sensitivity for all respiratory pathogens (74.83% vs. 33.11%, P < 0.001). Additionally, this method substantially reduced reporting time and cost only $150, which was one-fourth the cost of metagenomic next-generation sequencing(mNGS). This made tNGS particularly cost-effective for people who had mixed infections with Mycobacterium and other respiratory pathogens. Tafess et al. [45] utilized two commercial sequencing platforms to evaluate the comprehensive drug resistance of MTB cultures. The results demonstrated that both platforms achieved an average clinical sensitivity of 94.8% and a clinical specificity of 98.0%. Although the per-sample cost of the MinION platform (US $71.56) was marginally higher than that of the MiSeq

platform (US $67.83), the MinION platform offered a significantly shorter turnaround time, completing analysis 23 hours faster than the MiSeq platform (15 hours vs. 38 hours) and 18 days faster than pDST. This method could provide comprehensive drug-resistance information to patients with drug-resistant tuberculosis in a timely and cost-effective manner.

We propose that low- and middle-income countries bearing significant TB burden prioritize the use of tNGS for initial TB screening, particularly for rifampicin-resistant TB. Firstly, tNGS can facilitate real-time analysis of Mycobacterium tuberculosis DNA using portable devices such as the MinION, thereby minimizing reliance on high-level laboratory infrastructure (such as large and costly thermal cyclers [46]) and associated sample transportation costs compared to other detection methods. Secondly, in resource-limited environments, integrating Xpert MTB/RIF with tNGS can facilitate a more rapid and accurate diagnosis of patients with rifampicin-resistant tuberculosis (RR-TB) and multidrug-resistant tuberculosis (MDR-TB), avoiding unnecessary drug treatment costs and allowing for timely and effective treatment. Lastly, we must acknowledge that in low- and middle-income countries, traditional detection methods may be preferable to tNGS for tuberculosis diagnosis. However, tNGS is adequate to lessen a substantial financial burden on both society and the patients, even if it is used to provide laboratory evidence for TB patients with negative traditional detection results.

Our research had certain limitations. Initially, this study was retrospective, potentially leading to patient selection bias, and it was a study conducted at a single center with a small sample size, demanding the caution of generalizing research findings. Next, histopathological specimens used for extrapulmonary tuberculosis required a higher level of proficiency and expertise from operators than readily available respiratory specimens for pulmonary tuberculosis [47]. Inadequate sampling might compromise the accuracy of test results in this study. Finally, our study only reported tNGS results for detecting rifampicin-resistant mutations and did not fully elucidate its utility in detecting MTB resistance. Further research is urgently needed.

## 5. Conclusion

To sum up, tNGS is a versatile technology specialized in detecting low bacterial load clinical specimens. It has significant advantages in diagnosing low-bacterial-count tuberculosis (including extrapulmonary and smear-negative tuberculosis), identifying mycobacteria, and detecting drug-resistant mutations. In the future, tNGS is expected to be a key clinical component in the early prevention and management of tuberculosis.

## Supporting information

**S1 Data. Minimal data set.**
(XLSX)

## Author contributions

**Conceptualization:** Chen Yang.

**Formal analysis:** Chen Yang.

**Funding acquisition:** Yi Zeng.

**Methodology:** Chen Yang, Weiwei Gao.

**Project administration:** Yi Zeng.

**Resources:** Weiwei Gao.

**Software:** Chen Yang.

**Supervision:** Yi Zeng.

**Validation:** Weiwei Gao, Yicheng Guo.

**Writing – original draft:** Chen Yang.

**Writing – review & editing:** Weiwei Gao, Yicheng Guo.

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
