## [Decision Letter · Decision Letter 0]

19 Nov 2024

Dear Dr. Zeng,

Thank you for submitting your manuscript to PLOS ONE. After careful consideration, we feel that it has merit but does not fully meet PLOS ONE’s publication criteria as it currently stands. Therefore, we invite you to submit a revised version of the manuscript that addresses the points raised during the review process.

**Of note, the manuscript's structure and approach closely parallel those of Yang et al. (2023) in Tropical Medicine and Infectious Diseases. To distinguish itself and provide meaningful contribution to the field, the manuscript requires substantial revision to:**

**1. Better establish the study's rationale and unique contribution**

**2. Clearly explain how its findings advance current understanding of tNGS applications beyond existing literature**

**3. Consider expanding the scope to include the platform's capabilities in resistance profiling at least in terms of rifampicin resistance**

We look forward to receiving your revised manuscript.

Kind regards,

Guocan Yu

Academic Editor

PLOS ONE

**Journal Requirements:**

Nanjing Health Science and Technology Development Special Fund [grant number: M2021073]

3. In the online submission form, you indicated that The datasets used and/or analyzed during the current study are available from the corresponding author on reasonable request.

Reviewers' comments:

Reviewer's Responses to Questions

**Comments to the Author**

1. Is the manuscript technically sound, and do the data support the conclusions?

Reviewer #1: Yes

Reviewer #2: Partly

2. Has the statistical analysis been performed appropriately and rigorously?

Reviewer #1: Yes

Reviewer #2: Yes

3. Have the authors made all data underlying the findings in their manuscript fully available?

Reviewer #1: Yes

Reviewer #2: No

4. Is the manuscript presented in an intelligible fashion and written in standard English?

Reviewer #1: Yes

Reviewer #2: Yes

**Reviewer #1: ** This study is novel and the findings are significant. However, it would be more attractive, if the following details could be furnished.

1. The methodology section mentions IGRA as one of the diagnostic criteria. If so, the same was not included during data analysis. Moreover, IGRA is not a confirmatory test for TB as per WHO's recommendation and all IGRA positive patients should be subjected to other confirmatory tests

2. Role of IGRA and clinical features in screening the patients for TB in this study may be mentioned

3. Usefulness and practicality of this study in resource limited settings and low and middle income nations may be mentioned in terms of cost burden and how the early detection using tNGS could actually benefit the patients economically

4. Sociodemographic profiling of the study participants may be mentioned in a tabular format.

5. Details about the resistance patterns and detection of loci for resistant genes may be mentioned if observed. It would be great if the same could be associated with socio-demographic factors

**Reviewer #2:**  The manuscript clearly highlights the use of tNGS for MTB detection and its more superior performance compared to existing TB diagnostic methods. The manuscript could be improved by addressing the given points:-

1. The introduction would benefit from greater specificity and a clearer articulation of the study's objectives as they relate to the results presented.

2. Although this is a retrospective analysis, have the authors done a post-hoc power calculation performed to validate the statistical significance of the study's sample size?

3. The authors need to specify the sequence read threshold parameters used in tNGS for determining MTB presence/absence, and clarify whether factors such as sequencing depth influenced diagnostic classification. Including a detailed tNGS result decision framework would enhance reproducibility and reader understanding.

4. Consider including a subgroup analysis of MTB and NTM coinfection cases.

5. The authors need clarify the reference standard used for sensitivity calculations. Were all diagnostic tests compared against culture as the gold standard?

6. It would be valuable to note any cases where tNGS failed to detect MTB that was identified by other diagnostic methods.

7. Beyond reporting positive detection rates, consider including sensitivity and specificity analyses for both sputum-positive and sputum-negative samples to demonstrate the added value of tNGS.

8. The discussion section would be strengthened by comparing your findings with other studies that have employed tNGS for MTB diagnosis and resistance analysis

9. Given the recommendation in the discussion section for use in low- and middle-income countries (LMICs), please address the cost-effectiveness of using tNGS solely for diagnosis when its primary advantage lies in resistance detection.

10. While resistance analysis was noted as beyond the scope of this study, including a concordance analysis of rifampicin resistance detection between Xpert and tNGS would add valuable insight and strengthen the discussion

11. The limitations section should be expanded to address how technical aspects of tNGS might impact its diagnostic accuracy.

12.Minor considerations - The authors can review the manuscript for the consistent use of past tense throughout, as this is a retrospective analysis.

**Do you want your identity to be public for this peer review?** For information about this choice, including consent withdrawal, please see our Privacy Policy

Reviewer #1: **Yes: ** Dr Krithikaa Sekar

Reviewer #2: No

---

## [Author Response · Author response to Decision Letter 1]

7 Jan 2025

Editor:

The changes in our revised manuscript are highlighted in blue, and the locations of corrections or supplements are specifically indicated in the corresponding responses. Thank you for checking again.

We have adjusted the style of the article by searching for previous papers published in “PLOS ONE”. Please review again.

2. Please state what role the funders took in the study. If the funders had no role, please state: "The funders had no role in study design, data collection and analysis, decision to publish, or preparation of the manuscript.

We have provided additional explanations in the online system. Please double check.

We have presented all data used in this study in the NGDC Bioproject, accession number PRJCA034394, and corrected the statement of data availability(Page 21). Please double check.

4.PLOS requires an ORCID iD for the corresponding author in Editorial Manager on papers submitted after December 6th, 2016. Please ensure that you have an ORCID iD and that it is validated in Editorial Manager. To do this, go to ‘Update my Information’ (in the upper left-hand corner of the main menu), and click on the Fetch/Validate link next to the ORCID field. This will take you to the ORCID site and allow you to create a new iD or authenticate a pre-existing iD in Editorial Manager.

ORCID: 0009-0003-9871-238X

Thank you for checking again.

Reviewer 1

1.The methodology section mentions IGRA as one of the diagnostic criteria. If so, the same was not included during data analysis. Moreover, IGRA is not a confirmatory test for TB as per WHO's recommendation and all IGRA positive patients should be subjected to other confirmatory tests.

We are very sorry for causing you such confusion. The main purpose of this study is to study the diagnostic efficacy of different pathogenic detection methods for tuberculosis. Others, such as IGRA, radiological results, and patients' clinical symptoms, only assist clinicians to make final diagnosis based on the results of etiology, so we do not make specific analysis, but choose to include them in the patients' baseline table (Page 10-11 Table 1) for a descriptive analysis. Please review again.

2.Role of IGRA and clinical features in screening the patients for TB in this study may be mentioned.

We are very sorry for causing you such confusion. The main purpose of this study is to study the diagnostic efficacy of different pathogenic detection methods for tuberculosis. Others, such as IGRA, radiological results, and patients' clinical symptoms, only assist clinicians to make final diagnosis based on the results of etiology, so we do not make specific analysis, but choose to include them in the patients' baseline table(Page 10-11 Table 1) for a descriptive analysis. Please review again.

3. Usefulness and practicality of this study in resource limited settings and low and middle income nations may be mentioned in terms of cost burden and how the early detection using tNGS could actually benefit the patients economically。

Thank you for raising the feasibility and practicality of tNGS in resource limited environments and low and middle income countries. In the discussion section, we have supplemented the cost-effectiveness of tNGS in MTB diagnosis and the specific reasons for its application in low-income and high burden tuberculosis areas(Page 16 Line 29-Page 17 Line 14). Please double check.

4. Sociodemographic profiling of the study participants may be mentioned in a tabular format.

We are very sorry for causing you such confusion. The main purpose of this study is to study the diagnostic efficacy of different pathogenic detection methods for tuberculosis. Others, such as IGRA, radiological results, and patients' clinical symptoms, only assist clinicians to make final diagnosis based on the results of etiology, so we do not make specific analysis, but choose to include them in the patients' baseline table(Page 10-11 Table 1) for a descriptive analysis. Please review again.

5. Details about the resistance patterns and detection of loci for resistant genes may be mentioned if observed. It would be great if the same could be associated with socio-demographic factors

Thank you for your valuable insight on our manuscript. We have supplemented the consistency analysis of tNGS and Xpert on rifampicin resistance in the discussion section, and added the content of rifampicin resistant mutations discovered by tNGS, aiming to enrich the conclusions of the article(Page 20 Line 20- Page 21 Line 5). Thank you for checking again.

Reviewer 2

1.The introduction would benefit from greater specificity and a clearer articulation of the study's objectives as they relate to the results presented.

Thank you for your insightful query demanding our correction of the study’s objectives. We have used more specific and clearer language to express the research’s objective(Page 3 Line 29-Page 4 Line 13). Thank you for checking again.

2.Although this is a retrospective analysis, have the authors done a post-hoc power calculation performed to validate the statistical significance of the study's sample size?

We appreciate you bringing your concerns to our attention. Nevertheless, we contend that this research serves not only as a retrospective analysis but also as a diagnostic investigation. Therefore, efficacy analysis should not be employed to interpret historical clinical data. Furthermore, our objective is to accurately represent the diagnostic effectiveness of each detection method for a specific population over a defined timeframe. We acknowledge, however, that studies conducted at a single center may introduce selection bias and lead to certain distortions, which is recognized as one of the limitations of our study. Thank you for your understanding!

3.The authors need to specify the sequence read threshold parameters used in tNGS for determining MTB presence/absence, and clarify whether factors such as sequencing depth influenced diagnostic classification. Including a detailed tNGS result decision framework would enhance reproducibility and reader understanding.

Thank you for your insightful comments in our manuscript. We have addressed your concerns in the section of Material and methods( Page 8-9 2.7 Threshold parameters and decision framework for tNGS results). Plesase check again.

4.Consider including a subgroup analysis of MTB and NTM coinfection cases.

We appreciate your suggestions. However, according to our research design, patients with tuberculosis are classified in the positive category, whereas non-tuberculosis patients, including those with NTM infections, are categorized as negative. Our focus is solely on the differential diagnosis between these two categories using various testing methods. Including tuberculosis patients with NTM infections would hinder our ability to assess the diagnostic efficacy of each method. Consequently, we have decided to exclude these patients during the admission and exclusion phase. Thank you for your understanding.

5.The authors need clarify the reference standard used for sensitivity calculations. Were all diagnostic tests compared against culture as the gold standard?

We apologize for not specifying the benchmark utilized to assess the diagnostic performance of all detection methods, which impacted your review process. We have included the necessary explanations in the results section of the revised manuscript(Page 11 Line 4-7). Thank you for checking again.

6. It would be valuable to note any cases where tNGS failed to detect MTB that was identified by other diagnostic methods.

Thank you for your feedback to us. We have added this content to the discussion section(Page 18-Page 19). Thank you for checking again.

7. Beyond reporting positive detection rates, consider including sensitivity and specificity analyses for both sputum-positive and sputum-negative samples to demonstrate the added value of tNGS.

Thank you for your valuable feedback. Traditional pathogen testing negative pulmonary tuberculosis patients have always been the target of clinical doctors for early detection and treatment. We compared the positive detection rates of tNGS with other testing methods for smear-negative and culture-negative pulmonary tuberculosis patients through Tables 4 and 5(Page 14-Page 15), aiming to highlight the significant advantages of tNGS in this aspect. Thank you for checking again.

8. The discussion section would be strengthened by comparing your findings with other studies that have employed tNGS for MTB diagnosis and resistance analysis

Thank you for your insightful feedback on us. We have selected some representative references from recent years(such as reference 31 and 58) and compared them with our research results, aiming to improve the reliability of our research conclusions. Thank you for checking again.

9. Given the recommendation in the discussion section for use in low- and middle-income countries (LMICs), please address the cost-effectiveness of using tNGS solely for diagnosis when its primary advantage lies in resistance detection.

Thank you for raisingyour concern about the cost-effectiveness of using tNGS solely for diagnosis when its primary advantage lies in resistance detection. In the discussion section, we have supplemented the cost-effectiveness of tNGS in MTB diagnosis and the specific reasons for its application in low-income and high burden tuberculosis areas(Page 16 Line 29-Page 17 Line 14). Please double check.

10. While resistance analysis was noted as beyond the scope of this study, including a concordance analysis of rifampicin resistance detection between Xpert and tNGS would add valuable insight and strengthen the discussion

Thank you for your valuable insight on our manuscript. We have supplemented the consistency analysis of tNGS and Xpert on rifampicin resistance in the discussion section, and added the content of rifampicin resistant mutations discovered by tNGS, aiming to enrich the conclusions of the article(Page 20 Line 20- Page 21 Line 5). Thank you for checking again.

11. The limitations section should be expanded to address how technical aspects of tNGS might impact its diagnostic accuracy.

Thank you for your important feedback. We have added some possible reasons for errors in tNGS in the limitations section of our discussion (Page 21 Line 10-17). Thank you for checking again.

12.Minor considerations - The authors can review the manuscript for the consistent use of past tense throughout, as this is a retrospective analysis.

Thank you for your careful consideration. We have reviewed the manuscript for the consistent use of past tense throughout. Please check again.

---

## [Decision Letter · Decision Letter 1]

2 Feb 2025

Dear Dr. Zeng,

Thank you for submitting your manuscript to PLOS ONE. After careful consideration, we feel that it has merit but does not fully meet PLOS ONE’s publication criteria as it currently stands. Therefore, we invite you to submit a revised version of the manuscript that addresses the points raised during the review process.

We look forward to receiving your revised manuscript.

Kind regards,

Guocan Yu

Academic Editor

PLOS ONE

Journal Requirements:

Reviewers' comments:

Reviewer's Responses to Questions

**Comments to the Author**

Reviewer #1: All comments have been addressed

Reviewer #2: All comments have been addressed

2. Is the manuscript technically sound, and do the data support the conclusions?

Reviewer #1: Yes

Reviewer #2: Yes

3. Has the statistical analysis been performed appropriately and rigorously?

Reviewer #1: Yes

Reviewer #2: Yes

4. Have the authors made all data underlying the findings in their manuscript fully available?

Reviewer #1: Yes

Reviewer #2: Yes

5. Is the manuscript presented in an intelligible fashion and written in standard English?

Reviewer #1: Yes

Reviewer #2: Yes

Reviewer #1: I just wanted to thank the author for taking the feedback on a positive note and meticulously addressing each and every comment. TIts really glad to see such an impressive improvised manuscript and thank you for the efforts to revise it.

Reviewer #2: The authors have addressed all the comments and queries satisfactorily.

Minor comments.

The manuscript would benefit if the following points are considered:-

1. Shifting the comparison of rifampicin resistance patterns between tNGS and Xpert from discussion to result section with a concise mention of these comparative insights in broader diagnostic context in the discussion

2. In discussion section, comparing the cost effectiveness to other WGS platforms or culture DST and mentioning how authors envision cost reduction when using tNGS in LMICs

3. Comprehensive grammar review

**Do you want your identity to be public for this peer review?** For information about this choice, including consent withdrawal, please see our Privacy Policy

Reviewer #1: **Yes: ** Dr.Krithikaa Sekar

Reviewer #2: No

---

## [Author Response · Author response to Decision Letter 2]

28 Mar 2025

Dear Editor:

Thank you for your hard work in reviewing this research. We have uploaded the figure files involved in this study to the Preflight Analysis and Conversion Engine (PACE) digital diagnostic tool, account: 751289898@qq.com. Please check it.

Dear Reviewers:

The following are the responses to the reviewers. The changes in our revised manuscript are highlighted in blue, and the locations of corrections or supplements are specifically indicated in the corresponding responses. Thank you for your hard work in reviewing the manuscript.

1. Shifting the comparison of rifampicin resistance patterns between tNGS and Xpert from discussion to result section with a concise mention of these comparative insights in broader diagnostic context in the discussion.

Thank you for your insightful query. We have added 3.7 Evaluation of tNGS detection of rifampicin resistance patterns in the Results section of the article (Page 16 Line 5-Line 12). And the distribution of rifampicin-resistant mutations detected by tNGS on the rpoB gene was shown using a sunburst chart. In addition, in the discussion part of the article, we highlighted the advantages of tNGS in detecting rifampicin-resistant mutations outside RRDR compared with Xpert MTB/RIF, and the advantages of tNGS in providing comprehensive drug resistance information in the context of a wider range of drug-resistant tuberculosis (Page 20 Line 23-Page 21 Line 15). Please check again.

2. In discussion section, comparing the cost effectiveness to other WGS platforms or culture DST and mentioning how authors envision cost reduction when using tNGS in LMICs.

Thank you for your insightful advice. In the article's discussion section, we elaborated on the cost-effectiveness of tNGS compared to other detection methods for identifying mycobacteria and resistance mutations(Page 21 Line 16-Page 22 Line 6); additionally, we envision how to use tNGS to save costs in low- and middle-income countries(Page 22 Line 7-Line 19). Please check again.

3. Comprehensive grammar review

We have checked the article's grammar using Grammarly. Thank you for checking again.

---

## [Decision Letter · Decision Letter 2]

21 Apr 2025

Nanopore-based targeted next-generation sequencing (tNGS): a versatile technology specialized in detecting low bacterial load clinical specimens

PONE-D-24-36979R2

Dear Dr. Zeng,

We’re pleased to inform you that your manuscript has been judged scientifically suitable for publication and will be formally accepted for publication once it meets all outstanding technical requirements.

Kind regards,

Guocan Yu

Academic Editor

PLOS ONE

Additional Editor Comments (optional):

Reviewers' comments:

Reviewer's Responses to Questions

**Comments to the Author**

Reviewer #2: All comments have been addressed

2. Is the manuscript technically sound, and do the data support the conclusions?

Reviewer #2: Yes

3. Has the statistical analysis been performed appropriately and rigorously?

Reviewer #2: Yes

4. Have the authors made all data underlying the findings in their manuscript fully available?

Reviewer #2: Yes

5. Is the manuscript presented in an intelligible fashion and written in standard English?

Reviewer #2: Yes

Reviewer #2: (No Response)

**Do you want your identity to be public for this peer review?** For information about this choice, including consent withdrawal, please see our Privacy Policy

Reviewer #2: No

---

## [Editor Report · Acceptance letter]

PONE-D-24-36979R2

PLOS ONE

Dear Dr. Zeng,

I'm pleased to inform you that your manuscript has been deemed suitable for publication in PLOS ONE. Congratulations! Your manuscript is now being handed over to our production team.

Kind regards,

on behalf of

Dr. Guocan Yu

Academic Editor

PLOS ONE